# Weak, Broken, but Working—Intramolecular Hydrogen Bond in 2,2′-bipyridine

**DOI:** 10.3390/ijms241210390

**Published:** 2023-06-20

**Authors:** Ilya G. Shenderovich

**Affiliations:** Institute of Organic Chemistry, University of Regensburg, 93053 Regensburg, Germany; ilya.shenderovich@ur.de

**Keywords:** non-covalent interactions, NMR, proton transfer, 1,10-phenanthroline, GIAO

## Abstract

From an academic and practical point of view, it is desirable to be able to assess the possibility of the proton exchange of a given molecular system just by knowing the positions of the proton acceptor and the proton donor. This study addresses the difference between intramolecular hydrogen bonds in 2,2′-bipyridinium and 1,10-phenanthrolinium. Solid-state ^15^N NMR and model calculations show that these hydrogen bonds are weak; their energies are 25 kJ/mol and 15 kJ/mol, respectively. Neither these hydrogen bonds nor N-H stretches can be responsible for the fast reversible proton transfer observed for 2,2′-bipyridinium in a polar solvent down to 115 K. This process must have been caused by an external force, which was a fluctuating electric field present in the solution. However, these hydrogen bonds are the grain that tips the scales precisely because they are an integral part of a large system of interactions, including both intramolecular interactions and environmental influence.

## 1. Introduction

Non-covalent interactions make our world alive. Of particular importance, at least for biological systems, is hydrogen bonding. Among other things, this interaction is an almost indispensable initial step for the simplest molecular chemical reaction, and at the same time, the most important biological chemical reaction—the proton transfer reaction [1,2]. The rate of this reaction strongly depends on the distance between the proton acceptor and the proton donor groups, as well as on the deviation of the formed hydrogen bond from linearity. From an academic and practical point of view, it is therefore desirable to be able to assess the possibility of the proton exchange of a given molecular system with limited mobility just by knowing the position and mutual orientation of the proton acceptor and the proton donor. One of the ways to solve this problem is to study model systems for which the arrangement of these groups is similar but the characteristics of the hydrogen bond and associated proton exchange are different. 2,2′-bipyridine (BiPy) and 1,10-phenanthroline (Phen) represent a suitable model pair for such a study (Figure 1).

Theoretical calculations show that both the *cis* and *trans* conformers of BiPy can be present in the gas phase and solutions [3]. The *trans* conformer is lower in energy, and the barrier for *cis*/*trans* interconversion is small. This conformer is planar in the gas [3] and solid phases [4,5,6] but can be nonplanar in solutions [7]. When adsorbed onto silica, BiPy also adopts the *trans* conformation. [8]. In contrast, for 2,2′-bipyridinium (BiPyH^+^), its *cis* conformer is the low-energy one in the gas phase [3], solution [9], and the solid state [10,11,12,13,14,15,16]. This conformer is planar in the gas phase [3] and, as an exception, in the solid state in the case of non-coordinating anions [17,18,19,20,21,22,23,24,25]. The N…N distance in these structures varies from 2.611 Å [10] to 2.727 Å [23]. A detailed examination suggested that the N…N distance is determined more by the crystal packing than by the planarity of the molecule itself [25]. 

BiPy can be doubly protonated. In this form, its conformation tends back to the *trans* orientation, and the torsion angle between its rings varies from 36° to 180° [23,26].

The most stable conformation of other bipyridines often depends on intramolecular interactions [27,28]. For BiPyH^+^, the stabilization of the *cis* conformer is intuitively associated with the formation of an intramolecular hydrogen bond. In this regard, it is often compared with 1,10-phenanthrolinium (PhenH^+^). Indeed, the fundamental coordination [11] and proton-accepting properties of BiPy and Phen are similar. 

The N…N distances in Phen and PhenH^+^ are about 2.71–2.74 Å [10,11,14,16,17,23,29,30]. Phen can be doubly protonated. In this form, the N…N distance is about 2.80–2.90 Å [31,32,33,34,35,36], and the planarity of the molecule can be somewhat distorted. This allows us to make the following assumptions. The molecular structure of Phen is flexible enough to accommodate intermolecular non-covalent interactions. This is evident from the possibility of the double protonation and some contraction in PhenH^+^.

The difference between the intramolecular hydrogen bonds in BiPyH^+^ and PhenH^+^ has been addressed in the past. The following arguments were given as proof of the presence of a strong hydrogen bond in BiPyH^+^. The ν(NH) stretching frequency at 10 K is 3147 cm^−1^, while in PhenH^+^, it is 3279 cm^−1^ [10]. A lower limit on the strength of the intramolecular hydrogen bond in BiPyH^+^ was placed at 63 kJ/mol [3]. A fast reversible intramolecular hydrogen bond transfer in BiPyH^+^ was observed in a polar solvent down to 115 K [9]. This process is consistent with the presence of a strong intramolecular hydrogen bond. However, all these arguments only do not contradict the presence of this interaction, but they do not allow us to evaluate its strength either numerically or in comparison with the hydrogen bond in PhenH^+^. On the other hand, the difference between the N…N distances in BiPyH^+^ and PhenH^+^ is small. If we take into account that both bonds are nonlinear, then the difference between their energies can be insignificant.

This work has the following objectives: (i) to measure the ^15^N NMR chemical shifts of BiPy, Phen, BiPyH^+^, and PhenH^+^ in the solid state and to compare these experimental values with theoretically calculated values to determine the correct hydrogen bond geometry in the condensed state; (ii) to estimate the energy of the intramolecular hydrogen bonds in BiPyH^+^ and PhenH^+^; (iii) to theoretically model the intramolecular proton transfer in BiPyH^+^ and PhenH^+^ to elucidate the most likely mechanism and barrier height. 

## 2. Results and Discussion

### 2.1. Hydrogen Bond Geometry and Energy in BiPyH^+^ and PhenH^+^

#### 2.1.1. Experimental ^15^N NMR Chemical Shifts

Figure 2 shows the ^15^N NMR spectra of polycrystalline samples of BiPy, Phen, BiPyH^+^, and PhenH^+^. The numerical values of the isotropic chemical shift, $\delta_{iso}\left({}^{15}N\right)$, are collected in Table 1. The $\delta_{iso}\left({}^{15}N\right)$ values of BiPy and Phen were similar in the solid state and in a solution of tetrahydrofuran. There are two structurally distinct molecules in the crystal structure of Phen [29,30]. The anisotropy of the chemical shift tensors of structurally different molecules is different, which results in the difference in their $\delta_{iso}$ and spinning sideband patterns [37].

Hydrogen bonding significantly affects the $\delta_{iso}\left({}^{15}N\right)$ of pyridines [38]. For symmetrically substituted pyridines, there is a functional dependence between the change in the $\delta_{iso}\left({}^{15}N\right)$ and the value of the N…H distance [39]. This dependence was successfully used in the past to study the geometry of hydrogen bonds in complex molecular systems [40,41,42]. The maximal difference between the free and protonated forms is approximately the same for all pyridine derivatives and is 125 ppm [43]. 

The changes observed for BiPy and BiPyH^+^, as well as for Phen and PhenH^+^, are completely consistent with the expected changes (Table 1). The $\delta_{iso}\left({}^{15}N\right)$ of the protonated nitrogen shifted to −200 ppm, while for the other, it changed by only −10 ppm. However, neither for BiPyH^+^ nor for PhenH^+^ could these changes be converted into the numerical values of the N-H and H…N distances using the mentioned dependence. The reasons for this limitation are as follows. (i) This dependence was established for linear hydrogen bonds. In BiPyH^+^ and PhenH^+^, these bonds are highly bent. (ii) The conformations of BiPy and BiPyH^+^ are different. (iii) In PhenH^+^, the electron densities in the two conjugated pyridine rings cannot be considered independent. It is not known how these effects affect the values of the respective $\delta_{iso}\left({}^{15}N\right)$. The geometry of the hydrogen bonds in these structures will have to be determined using theoretical calculations.

#### 2.1.2. Calculated ^15^N NMR Chemical Shifts

Non-covalent interactions cause changes in chemical shifts [44,45]. For solids, the theoretical evaluation of these changes requires either using the GIPAW approach [46,47,48] or fragment-based calculations [46,49,50]. Such calculations require knowledge of the exact crystal structure, which is not always available [51]. However, in many cases, the influence of intermolecular interactions is small, and the required accuracy is achieved using simple adducts or single-molecule calculations [52,53,54]. 

Figure 3 shows the notations for the selected distances and angles used in this work. The calculated structures discussed below are named S_FB_W_ωM. S stands for the structure. F and B stand for the DFT functional and the basis set used for geometry optimization. F = ω or T for ωB97XD and TPSSh. B = t or q for def2tzvp and def2qzvp. W stands for geometries optimized with the polarizable continuum model approximation (PCM) using SCRF=(Solvent=water). ω and M stand for the DFT functional (ωB97XD) and the basis set used for GIAO NMR calculations, M = t or q or S for def2tzvp, def2qzvp, and pcSseg-3.

The structure BiPy_Tq_W_ωt was optimized using the TPSSh/def2qzvp approximation and SCRF=(Solvent=water). The $\delta_{iso}\left({}^{15}N\right)$ values were calculated using the ωB97XD/def2qzvp approximation and SCRF=(Solvent=water). The selected geometric and NMR parameters for this structure are reported in Table 2. The numerical values of these parameters depend on the approximation used for geometry optimization but not on the approximation used for the NMR calculations; see the structures BiPy_Tq_W_ωq, BiPy_ωq_W_ωt, and BiPy_ωq_W_ωq. The $\delta_{iso}\left({}^{15}N\right)$ values obtained for all these structures agree with the experimental values reported in Table 1. Therefore, the choice of the preferred approximation is a matter of taste.

BiPy*cis*_Tq_W_ωt corresponded to the *cis* form of BiPy, which is a higher-energy stable structure. The $\delta_{iso}\left({}^{15}N\right)$ of this structure is somewhat different. Therefore, the functional dependence of the
$\delta_{iso}\left({}^{15}N\right)$ on the N…H distance found for pyridines [39] cannot be used for BiPy since it is not clear whether the chemical shift of the *cis* or *trans* conformer corresponds to the limit N…H→∞.

BiPyH^+^_Tq_W_ωt is the optimized structure of BiPyH^+^. BiPyHClO_4__Tq_W_ωt is a structure in which the positions of all atoms, except for hydrogen atoms, were taken from the experimental XRD structure of BiPyH^+^ [ClO_4_]^−^, while the hydrogen positions were optimized using the TPSSh/def2qzvp approximation and SCRF=(Solvent=water). ZUTDAT_Tq_W_ωt and ZUTDAT01_Tq_W_ωt were constructed in the same way using the experimental XRD structure of BiPyH^+^ [B(C_6_H_5_)_4_]^−^ listed in the Cambridge Structural Database under ID ZUTDAT [10] and ZUTDAT01 [55]. A comparison of the $\delta_{iso}\left({}^{15}N\right)$ values for these structures suggests that they do not correlate unambiguously with the r_1_ and r_2_ distances. Of course, what is meant is not the obvious difference in the $\delta_{iso}\left(N^{p}\right)$ and $\delta_{iso}\left(N^{f}\right)$, but the variation in these values for different structures. These differences cannot be unambiguously attributed to the influence of the intramolecular hydrogen bond. Rather, they arose from differences in the geometries of these structures. On the other hand, protonation led to a significant decrease in the r_3_ compared to BiPy*cis*_Tq_W_ωt. 

It is not a priori obvious which basis set is required for the correct computation of the $\delta_{iso}\left({}^{15}N\right)$ of Phen. Three basis sets were used to calculate the $\delta_{iso}\left({}^{15}N\right)$ of Phen structures optimized using the TPSSh/def2qzvp approximation and SCRF=(Solvent=water): Phen_Tq_W_ωt, Phen_Tq_W_ωq, and Phen_Tq_W_ωS (Table 3). All the basis sets gave the same value. OPENAN_Tt_W_ωt and OPENAN01_Tt_W_ωt are structures in which the positions of all atoms, except for hydrogen atoms, were taken from the experimental XRD structures of Phen listed in the Cambridge Structural Database under ID OPENAN [29] and OPENAN01 [30], while the hydrogen positions were optimized using the TPSSh/def2qzvp approximation and SCRF=(Solvent=water). There are two structurally distinct molecules in these crystal structures that have different $\delta_{iso}\left({}^{15}N\right)\;\mathrm{values}$. The r_3_ distances in the solid state were less than the calculated one. The calculated values of the $\delta_{iso}\left({}^{15}N\right)$ approximately coincide, agree with the experimental values, and do not correlate with the r_3_. 

Protonation resulted in a shortening of the r_3_ distance. PhenH^+^_Tq_W_ωt corresponded to the optimized structure of PhenH^+^. ZUTDEX_Tq_W_ωt was constructed from the experimental XRD structure of PhenH^+^ [B(C_6_H_5_)_4_]^−^ listed in the Cambridge Structural Database under ID ZUTDEX [10] in the same manner as described above. The calculated values of $\delta_{iso}\left({}^{15}N\right)$ are similar and agree with the experimental values (Table 1). 

The first objective of this work was achieved. The experimental and calculated values of the $\delta_{iso}\left({}^{15}N\right)$ for BiPyH^+^ and PhenH^+^ agree quite well. Therefore, the calculated geometries of their intramolecular hydrogen bonds are reliable. Protonation resulted in a shortening of the r_3_ in both species. Although these distances are comparable to the shortest N…N distance measured for the proton-bound homodimers of the pyridines of 2.62 Å [56,57], the intramolecular hydrogen bonds in BiPyH^+^ and PhenH^+^ are strongly bent. Their lengths should be calculated as r_1_ + r_2_, which gives about 3.08 Å and 3.28 Å, respectively. The difference between the $\delta_{iso}\left({}^{15}N\right)$ in BiPy and Phen and the $\delta_{iso}\left({}^{15}N\right)$ of the N-H nitrogens in BiPyH^+^ and PhenH^+^ is greater than 120 ppm, the value expected for an uncoordinated pyridinium [43]. Therefore, both the geometries and $\delta_{iso}\left({}^{15}N\right)$ indicate that these hydrogen bonds are weak.

#### 2.1.3. Hydrogen Bond Energy

What is the energy of the intramolecular hydrogen bonds in BiPyH^+^ and PhenH^+^? Given the chemical similarity of these cations to the proton-bound homodimer of pyridine (PyHPy^+^), this energy can be estimated using simplified model calculations. Figure 4a shows the structure of PyHPy^+^. This complex has been studied in detail in the past [56,57,58,59,60,61,62,63,64,65]. Its hydrogen bond is moderately strong, it is of the asymmetric N-H…N type and exhibits fast reversible proton transfer in solution. Let us define the energy of the hydrogen bond in PyHPy^+^, ΔE, as the difference between the electronic energy of PyHPy^+^ for a given distance, r_2_, and the sum of the electronic energies of pyridine and pyridinium. At the ωB97XD/def2tzvp approximation and SCRF=(Solvent=water), ΔE is minimal at r_2_ = 1.645433 Å. This distance is longer than the experimental value of 1.532 Å [57]. The reason for this has been explained elsewhere [66,67]. As the r_2_ increases, the ΔE increases and tends asymptotically to zero (Figure 4b). The numerical values are collected in Appendix A in the Appendix A. 

For a large r_2_, one can expect that the binding energy of the hydrogen-bonded pyridine-N^p^-H…N^f^-pyridine cation is defined by the electric force between the charges located at the N^f^ nitrogen and the binding proton. As an approximate numerical estimate of these charges, one can use the Mulliken charges of the N^f^ nitrogen *q*(*N^f^*) and the binding proton *q*(*H*)。 The electric potential energy associated with this interaction is $\varepsilon =q\left(H\right)q\left(N^{f}\right)/r_{2}$, where *q*(*H*) and *q*(*N^f^*) are in units of the elementary charge *e*. Figure 4c shows *ε* as a function of the Δ*E*. For Δ*E* values between −41.6 kJ/mol and −8.5 kJ/mol, i.e., for r_2_ values between 1.95 Å and 3.0 Å, a linear dependence between the *ε* and Δ*E* is evident. Thus, for this interval of the r_2_ distances, the above assumption is correct. This dependence is $\Delta E=1730\cdot \varepsilon \left(\omega B97\mathrm{XD}/\mathrm{def}2\mathrm{tzvp}\right)+50$, where the Δ*E* is in kJ/mol and the *ε* in *e*^2^/Å (Figure 4d). (Of course, the energy of such hydrogen bonds can alternatively be defined as ε. In this case, however, there is ambiguity in the choice of the value of the relative permittivity, which requires additional assumptions that are difficult to justify.)

For the other approximations, the linearity of the dependence was preserved, but the numerical values of the coefficients changed. For the TPSSh/def2qzvp approximation, $\Delta E=2720\cdot \varepsilon \left(\mathrm{TPSSh}/\mathrm{def}2\mathrm{qzvp}\right)+98$ (Appendix A). For the ωB97XD/def2qzvp approximation, $\Delta E=2625\cdot \varepsilon \left(\omega B97\mathrm{XD}/\mathrm{def}2\mathrm{qzvp}\right)+107$ (Appendix A).

In BiPyH^+^ and PhenH^+^, the r_2_ was in the range pf 2.0–2.3 Å. Thus, the obtained dependences could be used to estimate the hydrogen bond energy in these cations. The numerical values of Δ*E* varied depending on the basis set but were the same for the TPSSh and ωB97XD DFT functionals (Table 4). Consequently, the energy values of the intramolecular hydrogen bonds in BiPyH^+^ and PhenH^+^ were about 25 kJ/mol and 15 kJ/mol, respectively. 

### 2.2. Intramolecular Proton Transfer in BiPyH^+^ and PhenH^+^

#### Proton Transfer Pathway

This section discusses the theoretically modeled intramolecular proton transfer pathway in BiPyH^+^ and PhenH^+^. Since the exact numerical values of the considered geometric and energetic parameters are not the subject of this analysis, these calculations were carried out without using the PCM approximation. The proton transfer pathway was modeled as follows. The geometries of BiPyH^+^ and PhenH^+^ were optimized using the ωB97XD/def2tzvp approximation. Then, the r_1_ distance was increased stepwise, and for each of the selected r_1_ distances, the geometry was optimized again. The final structure, referred to below as a symmetric structure, corresponds to equal r_1_ and r_2_ distances. Thus, this model describes the case of proton transfer due to the gradual strengthening of the intramolecular hydrogen bond, in which the geometry of the molecule has time to adapt to this change. The resulting numerical values are reported in Appendix A. 

Figure 5 shows the changes in the selected geometric parameters caused by such proton transfer. These changes are presented as functions of the proton coordinate q_1_ = ½(r_1_ − r_2_). The decrease in q_1_ caused a contraction of r_3_ (Figure 5a). The trends of these changes were the same for BiPyH^+^ and PhenH^+^, although, in the latter, the r_3_ was always longer by at least 0.05 Å. The changes in the angles α, β, and γ were the same as well (Figure 5b). Strengthening the hydrogen bond made it more linear, that is, the angle α increased. The convergence of the angles β and γ for a flexible BiPyH^+^ occurred smoothly. On the contrary, for a rigid PhenH^+^, the last step to q_1_ = 0 caused a saltatory change in these angles.

Figure 6 shows the changes in the heavy atom hydrogen bond coordinate q_2_ = r_1_ + r_2_ and the energy values of BiPyH^+^ and PhenH^+^ caused by such a proton transfer. As with any other known hydrogen bonds, the decrease in q_1_ caused a contraction of q_2_ [68,69,70] (Figure 6a). However, there was also a difference between the BiPyH^+^ and PhenH^+^. For a large q_1_, the q_2_ changed abnormally in PhenH^+^ and somewhat increased with a decreasing q_1_. This fact can be interpreted as an indication that, near the optimal geometry, the intramolecular hydrogen bond in PhenH^+^ is weak and small changes in its geometry do not significantly affect the geometry of this rigid structure. On the contrary, in the limit of the strong hydrogen bonding at q_1_ = 0, this bond was only slightly weaker than in BiPyH^+^. Near the optimal geometry, a small increase in the r_1_ did not cause a significant energy increase (Figure 6b). However, then the energy rapidly increased in an inverse proportion to q_1_. For PhenH^+^, this growth was much steeper than for BiPyH^+^. As a result, the energies of the symmetric structures at q_1_ = 0 were 41 kJ/mol and 72 kJ/mol higher than the energies of the lowest energy structures of BiPyH^+^ and PhenH^+^, respectively.

The energy required for the proton transfer, calculated above, is quite large. How much will this energy change if the polarity of the solvent is taken into account, SCRF=(Solvent=water)? The geometries of the lowest energy structures and the symmetric structures at q_1_ = 0 of BiPyH^+^ and PhenH^+^ changed somewhat in this approximation (Appendix A). Their energy differences increased to 51 kJ/mol and 82 kJ/mol for BiPyH^+^ and PhenH^+^, respectively.

To obtain a more realistic estimate of the energy required for the proton transfer, the calculation procedure was changed. The geometries were optimized using the ωB97XD/def2tzvp approximation and SCRF=(Solvent=water). Then, the r_3_ distance was decreased stepwise, and for each of the selected r_3_ distances, the geometry was optimized. The final structure corresponded to equal r_1_ and r_2_ distances. For all these structures, the N^p^-H stretches were calculated (Appendix A). Thus, this model describes the case of proton transfer in which the movement of the proton is much faster than the ongoing changes in the r_3_ distance and reduces the multi-dimensional potential energy problem [71,72,73] to a one-dimensional one. The only structure with a negative frequency is the structure with the equal r_1_ and r_2_ distances. Thus, these calculations show that no intramolecular vibrations can be the cause of the fast reversible proton transfer observed in BiPyH^+^ in the experiment [9]. This process must be caused by external forces. 

## 3. Materials and Methods

Solid-state ^15^N NMR measurements were performed on an *Infinity_plus_* spectrometer system (Varian Inc., Palo Alto, CA, USA) operated at 7 T and equipped with a Chemagnetics 6 mm pencil CPMAS probe. The ^15^N{^1^H} MAS CP NMR spectra were recorded using a 90°−pulse length of 5.0 μsec, a cross-polarization contact time of 15 ms, and relaxation delays of 5–20 s. The spectra were indirectly referenced to CH_3_NO_2_ [74] using solid ^15^NH_4_Cl (δ = −341.3 ppm [75]). To convert these values to the liquid ammonia scale, add 380.6 ppm to them [75].

The Gaussian 09.D.01 program package was used for geometry optimizations and NMR calculations [76]. The TPSSh and ωB97XD DFT hybrid functionals [77,78] and the def2tzvp, def2qzvp, and pcSseg-3 basis sets were used [79,80,81]. The geometry optimization was performed at the very tight convergence criteria. The NMR calculations were carried out using the GIAO approach. Some calculations were performed using the PCM approximation with water as the solvent [82]. This choice is arbitrary. The outcomes of the PCM approximation are not very sensitive to the value of the dielectric constant [83]. However, this correction is necessary to consider the effect of the crystal field. 

To convert the ^15^N NMR absolute shielding values σ obtained in the theoretical calculations into the chemical shift scale *δ* used in the experiments, it was necessary to know the reference ^15^N absolute chemical shielding σ^ref^: $\delta \approx \sigma^{ref}-\sigma$ [84,85]. The numerical value of σ^ref^ depends on the approximation used to calculate *σ* [86]. The calculated chemical shifts reported in this work were derived from the calculated absolute shielding values using the following values of σ^ref^: $\sigma^{ref}\left(\omega B97\mathrm{XD}/\mathrm{def}2\mathrm{tzvp}\right)=$ 143 ppm, $\sigma^{ref}\left(\omega B97\mathrm{XD}/\mathrm{def}2\mathrm{qzvp}\right)=$ 148 ppm, and $\sigma^{ref}\left(\omega B97\mathrm{XD}/\mathrm{pcSseg}3\right)=$ 153 ppm [86]. The applicability of this approach was previously confirmed for ^31^P NMR [54,87,88].

Crystalline BiPyH^+^ [ClO_4_]^−^ was obtained from a solution of 2,2′-bipyridine in tetrahydrofuran by adding 70% HClO_4_. The precipitate was filtered off and recrystallized either from acetonitrile or from nitromethane. X−ray diffraction data for single crystals of BiPyH^+^ [ClO_4_]^−^ were collected by Rigaku Oxford Diffraction SuperNova diffractometers (Appendix A, Applied Rigaku Technologies, Inc., Austin, TX, USA). The crystals were kept at 123.0 K during the data collection. The structures were solved with the ShelXT 2018/2 [89] solution program using dual methods and Olex2 1.5-alpha [90] as the graphical interface. The model was refined with Olex2.refine 1.5-alpha [91] using full matrix least squares minimization on ***F^2^***. The structures have been deposited at the Cambridge Crystallographic Data Centre (CCDC) with numbers 2266914 and 2266916. The atomic coordinates of the BiPyHClO4_Tq_W, ZUTDAT_Tq_W, ZUTDAT01_Tq_W, OPENAN_Tt_W, OPENAN01_Tt_W, and ZUTDEX_Tq_W structures are available in Appendix A.

Polycrystalline BiPyH^+^ [B(C_6_H_5_)_4_]^−^ and PhenH^+^ [B(C_6_H_5_)_4_]^−^ were obtained from a solution of BiPy or Phen in methanol by adding 37% HCl and NaB(C_6_H_5_)_4_. The precipitate was filtered off and washed with water.

## 4. Conclusions

This study had three objectives. First, it aimed to measure the ^15^N NMR chemical shifts of BiPy, Phen, BiPyH^+^, and PhenH^+^ in the solid state and to compare these experimental values with theoretically calculated values to determine the correct hydrogen bond geometry in the condensed state. These measurements and calculations show equal results. Thus, the calculated geometries were similar to those of the solid state. 

The second objective was to estimate the energy of the intramolecular hydrogen bond in BiPyH^+^ and PhenH^+^. The model calculations provided for the energy of these interactions were 25 kJ/mol and 15 kJ/mol, respectively. The nature of these hydrogen bonds is purely electrostatic. It would be instructive to make an independent estimate of these energies using the experimental electron density distribution function [92,93,94,95,96].

The final objective was to model the intramolecular proton transfer in BiPyH^+^ and PhenH^+^. The analysis showed that neither these interactions nor N-H stretches could be responsible for the fast reversible proton transfer observed for BiPyH^+^ in a polar solvent down to 115 K. The electrostatic interaction between the mobile proton and the second nitrogen atom was a guide for the transfer, but it was too weak to overcome the transfer barrier. This process was controlled by an external force. This force was a fluctuating electric field present in the solution. Its effects and amplitude have been evaluated in the past for several molecular adducts [66,83,88,97,98]. However, the influence of this field on proton transfer can only be modeled if the transfer occurs along the symmetry axis of the adduct under consideration. Neither BiPyH^+^ nor PhenH^+^ has such an axis. 

The particular results obtained for the examples of BiPyH^+^ and PhenH^+^ allow us to draw the following general conclusion. The hydrogen bonds under discussion are weak. By themselves, they could not have a significant effect on the properties of these molecules. However, these bonds are an integral part of a large system of interactions, which includes both intramolecular interactions and the influence of the environment. Within the framework of such a system, they become the grain that tips the scales. 

## Figures and Tables

**Figure 1 ijms-24-10390-f001:**
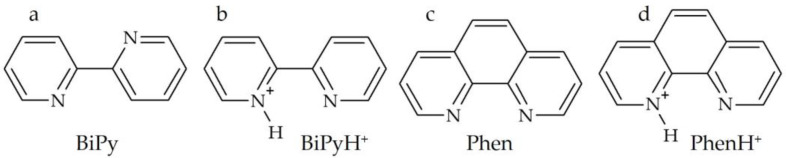
Substances studied in this work. (**a**) 2,2′-bipyridine (BiPy), (**b**) 2,2′-bipyridinium (BiPyH^+^), (**c**) 1,10-phenanthroline (Phen), (**d**) 1,10-phenanthrolinium (PhenH^+^).

**Figure 2 ijms-24-10390-f002:**
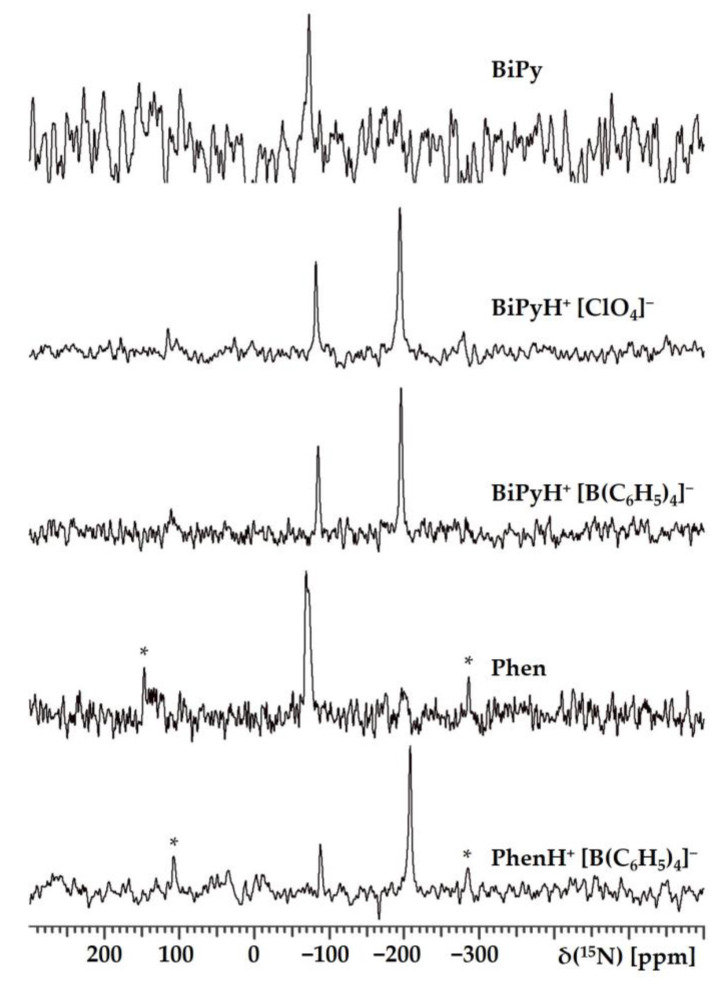
^5^N CPMAS NMR spectra of polycrystalline samples of BiPy, Phen, BiPyH^+^, and PhenH^+^. Asterisks * denote spinning sidebands.

**Figure 3 ijms-24-10390-f003:**
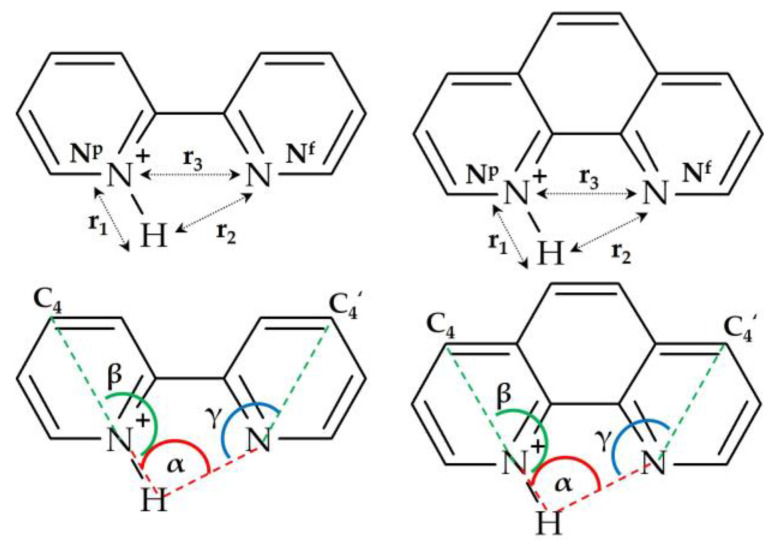
The notations for the selected distances and angles used in this work.

**Figure 4 ijms-24-10390-f004:**
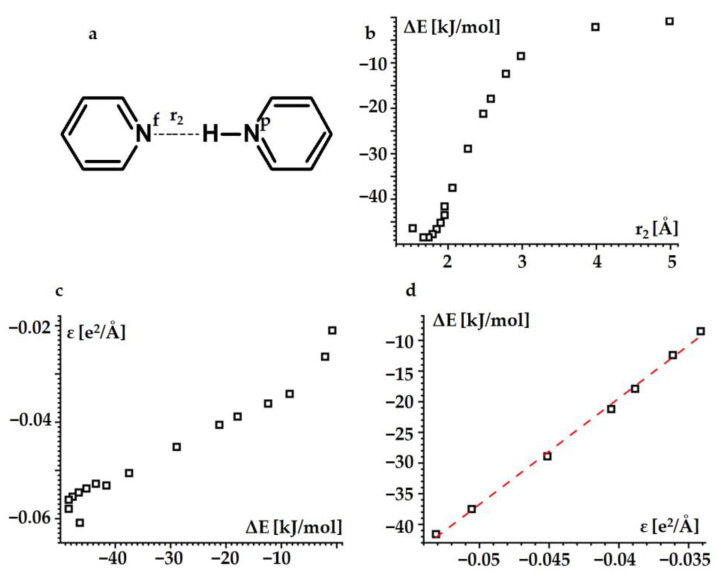
The hydrogen bond energy of PyHPy^+^ estimated at the ωB97XD/def2tzvp approximation and SCRF=(Solvent=water). (**a**) The structure of PyHPy^+^. (**b**) The hydrogen bond energy Δ*E* as a function of r_2_. (**c**) The electric potential energy *ε* associated with the N^f^…H interaction as a function of Δ*E*. (**d**) The range of the linear dependence between Δ*E* and *ε*.

**Figure 5 ijms-24-10390-f005:**
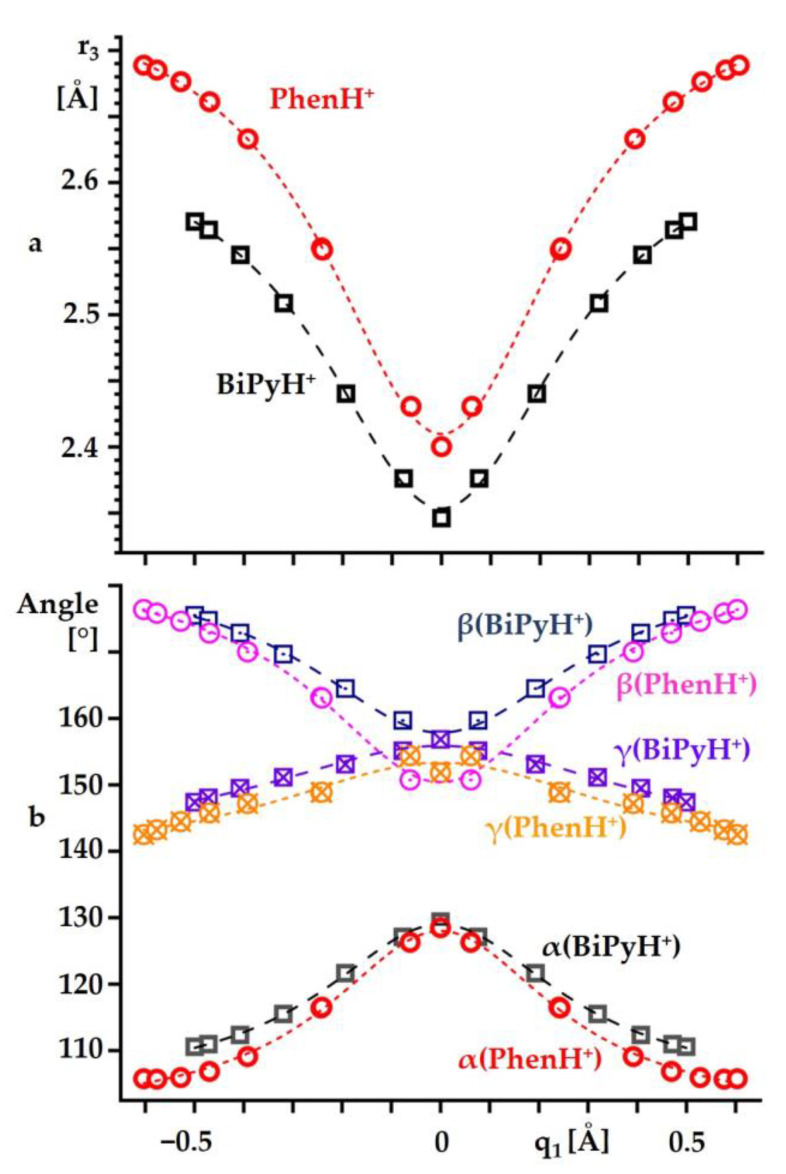
The changes in the r_3_ (**a**) and selected angles (**b**) caused by the intramolecular proton transfer in BiPyH^+^ and PhenH^+^ as functions of the proton coordinate q_1_ = ½(r_1_ − r_2_). Curves describe a trend and serve as a guide for the eyes.

**Figure 6 ijms-24-10390-f006:**
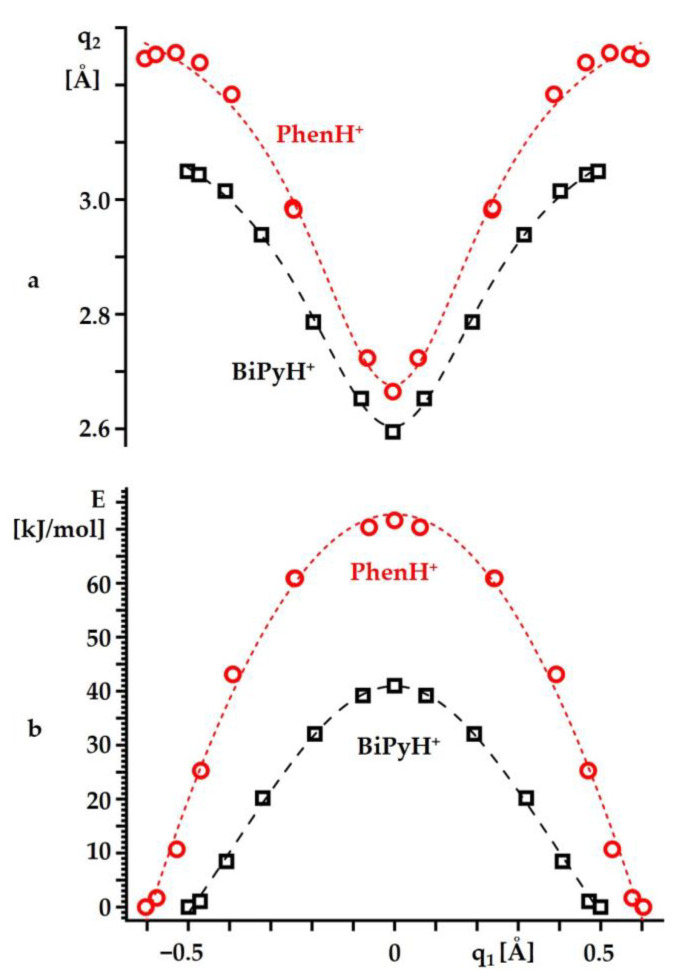
The changes in the heavy atom hydrogen bond coordinate q_2_ = r_1_ + r_2_ (**a**) and the energy (**b**) caused by the proton transfer in BiPyH^+^ and PhenH^+^ as functions of q_1_ = ½(r_1_ − r_2_). Curves describe a trend and serve as a guide for the eyes.

**Table 1 ijms-24-10390-t001:** The experimental values of the isotropic ^15^N chemical shift.

Substance	State	$\delta_{iso}$, ppm
BiPy	Polycrystalline	−72.7
BiPy	In tetrahydrofuran	−71.65
BiPyH^+^ [ClO_4_]^−^	Polycrystalline	−82.0 & −193.4
BiPyH^+^ [B(C_6_H_5_)_4_]^−^	Polycrystalline	−85.0 & −195.5
Phen	Polycrystalline	−68.6 & −72.2
Phen	In tetrahydrofuran	−66.74
PhenH^+^ [B(C_6_H_5_)_4_]^−^	Polycrystalline	−88.3 & −208.3

**Table 2 ijms-24-10390-t002:** Calculated geometries and $\delta_{iso}\left({}^{15}N\right)$ for the selected structures of BiPy and BiPyH^+^.

Structure	r_1_, Å	r_2_, Å	r_3_, Å	φ ^1^, °	$\delta_{iso}$, ppm
BiPy_Tq_W_ωt	−	−	3.61234	180	−70
BiPy_Tq_W_ωq	−	−	3.61234	180	−70
BiPy_ωq_W_ωt	−	−	3.60001	180	−75
BiPy_ωq_W_ωq	−	−	3.60001	180	−74
BiPy*cis*_Tq_W_ωt	−	−	2.72141	0	−63
BiPyH^+^_Tq_W_ωt	1.02483	2.05149	2.58497	0	−82 & −199
BiPyHClO_4__Tq_W_ωt	1.02871	2.24853	2.66167	14.6	−71 & −195
ZUTDAT_Tq_W_ωt	1.02365	2.08824	2.61085	4.3	−76 & −198
ZUTDAT01_Tq_W_ωt	1.02376	2.08657	2.60115	4.7	−82 & −202

^1^ The dihedral angle between pyridine rings.

**Table 3 ijms-24-10390-t003:** Calculated geometries and $\delta_{iso}\left({}^{15}N\right)$ for the selected structures of Phen and PhenH^+^.

Structure	r_1_, Å	r_2_, Å	r_3_, Å	$\delta_{iso}$, ppm
Phen_Tq_W_ωt	−	−	2.77022	−67
Phen_Tq_W_ωq	−	−	2.77022	−68
Phen_Tq_W_ωS	−	−	2.77022	−67
OPENAN_Tt_W_ωt	−	−	2.72215 2.72614	−65 −73
OPENAN01_Tt_W_ωt	−	−	2.73995 2.74499	−65 −63
PhenH^+^_Tq_W_ωt	1.02021	2.26272	2.70884	−84 & −207
ZUTDEX_Tq_W_ωt	1.01968	2.26295	2.70682	−81 & −204

**Table 4 ijms-24-10390-t004:** The r_2_ and r_3_ distances, the Mulliken charges of the binding proton (*q*(*H*)) and the N^f^ nitrogen (*q*(*N^f^*)), the electric potential energy ε, and the hydrogen bond energy Δ*E* in BiPyH^+^ and PhenH^+^ estimated at various approximations and SCRF=(Solvent=water).

Substance	r_2_, Å	r_3_, Å	*q*(*H*), *e*	*q*(*N^f^*), *e*	*ε* ^1^, *e*^2^/Å	Δ*E*, kJ/mol
BiPyH^+^_wt_W	2.079695	2.595704	0.293915	−0.394282	−0.0557	−46 ± 3
BiPyH^+^_Tq_W	2.051480	2.584967	0.255385	−0.362354	−0.0451	−25 ± 8
BiPyH^+^_wq_W	2.080350	2.595114	0.237234	−0.446997	−0.0510	−27 ± 4
PhenH^+^_wt_W	2.226297	2.688838	0.288186	−0.390619	−0.0506	−37 ± 3
PhenH^+^_Tq_W	2.262717	2.708843	0.259246	−0.351045	−0.0402	−11 ± 8
PhenH^+^_wq_W	2.273517	2.710015	0.246195	−0.424726	−0.0460	−14 ± 4

^1^ The electric potential energy ε=qHqNfr2 associated with the N^f^…H interaction.

## Data Availability

Not applicable.

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
