# Peer review of "Weak, Broken, but Working—Intramolecular Hydrogen Bond in 2,2′-bipyridine"

_ijms, 2023, doi:10.3390/ijms241210390_

Round 1

Reviewer 1 Report

This manuscript studied difference between intramolecular hydrogen bonds in 2,2’-bipyridinium and 1,10-phenanthrolinium by 15N NMR, estimation the energy of the intramolecular hydrogen bond and the intramolecular proton transfer. In that example, the author wanted to achieve assess the possibility of proton exchange for a given molecular system, by knowing the positions of the proton acceptor and the proton donor. But there are some issues that need to be addressed before it is considered for publication.

 1. In the 121 and 199 lines, the def2tzv and def2qzv lack polarization. This is inconsistent with the description at the end of the article and in the supporting materials. The lack of polarization makes it impossible to accurately describe the shape and nature of the electron cloud. The error in key length and vibration frequency is relatively large.

2. In the 240 line, you mentioned all calculations were carried out without using the PCM approximation. But I think the PCM solvent model must be added in this example, because the ionic system in the solution is involved, and the hydrogen ion (proton) needs to be free, so not adding the solvent model may lead to qualitative errors in structural optimization and electronic structure calculation.

Author Response

Reviewer 1.

This manuscript studied difference between intramolecular hydrogen bonds in 2,2’-bipyridinium and 1,10-phenanthrolinium by 15N NMR, estimation the energy of the intramolecular hydrogen bond and the intramolecular proton transfer. In that example, the author wanted to achieve assess the possibility of proton exchange for a given molecular system, by knowing the positions of the proton acceptor and the proton donor. But there are some issues that need to be addressed before it is considered for publication.

Answer: I appreciate the Reviewer’s comment and have addressed all the suggestions.

  1. In the 121 and 199 lines, the def2tzv and def2qzv lack polarization. This is inconsistent with the description at the end of the article and in the supporting materials. The lack of polarization makes it impossible to accurately describe the shape and nature of the electron cloud. The error in key length and vibration frequency is relatively large.

Answer: The Reviewer is right. I apologize for my carelessness. All calculations were performed using the def2tzvp and def2qzvp basis sets as specified in the Supporting Information. The typo has been corrected.

  1. In the 240 line, you mentioned all calculations were carried out without using the PCM approximation. But I think the PCM solvent model must be added in this example, because the ionic system in the solution is involved, and the hydrogen ion (proton) needs to be free, so not adding the solvent model may lead to qualitative errors in structural optimization and electronic structure calculation.

Answer: I completely agree that the use of the PCM approximation is mandatory for a correct assessment of the numerical values of the properties of molecular systems in a condensed state. However, as stated in the manuscript, the exact numerical values of the considered geometric and energetic parameters were not the subject of the section 2.2.1 Proton transfer pathway. Only these specific calculations were carried out without using the PCM approximation. Nevertheless, to eliminate ambiguity, I explicitly specified in the text that this approach is used only in this section: “Since the exact numerical values of the considered geometric and energetic parameters are not the subject of this analysis, these calculations were carried out without using the PCM approximation.”

Reviewer 2 Report

The manuscript "Weak, broken, but working − intramolecular hydrogen bond in 2,2’-bipyridine" focuses on 15N NMR and theoretical study of 2,2'-dipyridine and 1,10-phenanthroline. Chemical shifts deteermined experimentally are in good agreement with those calculated by GIAO method at any theory level exploited. The energies of intramolecular hydrogen bonds were estimated. The pathway of intramolecular hydrogen transfer from one nitrogen to another was drawn using the calculated values of key interatomic distances and angles. In general, paper is very interesting and contributes to the field of H-bonds studying. The experiment is quite non-routine and labor-costly. However, I still want to niggle around a bit:

1. Lines 179-181. The statement from the second sentence ("Therefore, the calculated geometries of their intramolecular hydrogen bonds are reliable.") is too bold to me. Actually, the agreement between experimental and calculated spectra means that the satisfatory level of theory was employed to predict the chemical shifts, and that's it (although the results achieved do not contradict a hypothesis of reliable geometry parameters either).

2. I feel that the theoretical study of intramolecular H-bonds can be performed in a more advanced way. Perhaps, AIM study is in order, which would allow for analysing bonds critical points, Bader's charges (seem better to me than Mulliken's ones) etc.

3. Dependencies on Fig. 4b,c look like logistic curves to me. Perhaps, it would be more corect to use the corresponding equation than a linear regression.

4. From my point of view, it seemed evident enough that the intramolecular transfer of proton in polar solvent most cases would require a molecule of polar solvent. In a right place and at a right time it could serve as a transmission link. Therefore, it could be of interest to try taking the solvent into account directly, not as a polarized continuum. A single molecule of, say, water introduced close to the site of proton transfer may decrease the calculated potential barrier drastically.

5. There are some typos (e.g., line 338, "15 kJ7mol").

6. I used to see that the optimized geometries of the structures under study are provided together with the manuscript as a bunch of xyz files or Cartesian coordinates tables in ESI.

The chicanery above do not diminish in any way the paper's virtues and could be cosidered just a possible way of further improvements.

Author Response

Reviewer 2.

The manuscript "Weak, broken, but working − intramolecular hydrogen bond in 2,2’-bipyridine" focuses on 15N NMR and theoretical study of 2,2'-dipyridine and 1,10-phenanthroline. Chemical shifts deteermined experimentally are in good agreement with those calculated by GIAO method at any theory level exploited. The energies of intramolecular hydrogen bonds were estimated. The pathway of intramolecular hydrogen transfer from one nitrogen to another was drawn using the calculated values of key interatomic distances and angles. In general, paper is very interesting and contributes to the field of H-bonds studying. The experiment is quite non-routine and labor-costly. However, I still want to niggle around a bit:

Answer: I appreciate the Reviewer's positive feedback.

  1. Lines 179-181. The statement from the second sentence ("Therefore, the calculated geometries of their intramolecular hydrogen bonds are reliable.") is too bold to me. Actually, the agreement between experimental and calculated spectra means that the satisfatory level of theory was employed to predict the chemical shifts, and that's it (although the results achieved do not contradict a hypothesis of reliable geometry parameters either).

Answer: Absolute shielding values calculated for the molecules under study were converted to the chemical shift scale using a reference absolute shielding obtained from independent calculations of other molecular systems. The resulting chemical shifts agree with the corresponding experimental values. The chemical shifts discussed are highly dependent on molecular geometry. Thus, the coincidence of the calculated and experimental chemical shifts confirms not only the adequacy of the approximation used in these calculations, but also the closeness of the calculated and real geometries of the considered molecular system.

  1. I feel that the theoretical study of intramolecular H-bonds can be performed in a more advanced way. Perhaps, AIM study is in order, which would allow for analysing bonds critical points, Bader's charges (seem better to me than Mulliken's ones) etc.

Answer: The Reviewer is right. There are many other approaches that can be used to estimate the energies of the hydrogen bonds under consideration. The AIM approach is of particular interest because it is applicable to both theoretical and experimental analysis. Therefore, it has been mentioned in the conclusion, refs. 92-96. Since at this stage I did not have any experimental estimates of the interaction energy, I limited myself to the simplest theoretical approach.

  1. Dependencies on Fig. 4b,c look like logistic curves to me. Perhaps, it would be more corect to use the corresponding equation than a linear regression.

Answer: The Reviewer is right. However, I deliberately did not analyze the behavior of the functional dependency over the entire distance interval. I was only interested in the interval in which this dependence is linear, since only in this interval the nature of these hydrogen bonds is purely electrostatic, Fig. 4d.

  1. From my point of view, it seemed evident enough that the intramolecular transfer of proton in polar solvent most cases would require a molecule of polar solvent. In a right place and at a right time it could serve as a transmission link. Therefore, it could be of interest to try taking the solvent into account directly, not as a polarized continuum. A single molecule of, say, water introduced close to the site of proton transfer may decrease the calculated potential barrier drastically.

Answer: I fully agree that such an analysis would be interesting. I doubt that a model with one or two solvent molecules can be instructive. But a molecular dynamics study would be very useful. However, such a study is beyond the scope of this work.

  1. There are some typos (e.g., line 338, "15 kJ7mol").

Answer: The typo has been corrected.

  1. I used to see that the optimized geometries of the structures under study are provided together with the manuscript as a bunch of xyz files or Cartesian coordinates tables in ESI.

The chicanery above do not diminish in any way the paper's virtues and could be cosidered just a possible way of further improvements.

Answer: The atomic coordinates of BiPyHClO4_Tq_W, ZUTDAT_Tq_W, ZUTDAT01_Tq_W, OPENAN_Tt_W, OPENAN01_Tt_W, and ZUTDEX_Tq_W have been included in the Supporting Information, Tables S11-S16.